# Retrosynthesis Prediction with Conditional Graph Logic Network

**Hanjun Dai**[‡†][*] **Chengtao Li**[□]**, Connor W. Coley**[◇]**, Bo Dai**[‡]**, Le Song**[†◦]
[‡]Google Research, Brain Team, {hadai, bodai}@google.com
[□]Galixir Inc., chengtao.li@galixir.com
[◇]Massachusetts Institute of Technology, ccoley@mit.edu
[†]Georgia Institute of Technology, [◦]Ant Financial, lsong@cc.gatech.edu

## Abstract

Retrosynthesis is one of the fundamental problems in organic chemistry. The task is to identify reactants that can be used to synthesize a specified product molecule. Recently, computer-aided retrosynthesis is finding renewed interest from both chemistry and computer science communities. Most existing approaches rely on template-based models that define subgraph matching rules, but whether or not a chemical reaction can proceed is not defined by hard decision rules. In this work, we propose a new approach to this task using the Conditional Graph Logic Network, a conditional graphical model built upon graph neural networks that learns when rules from reaction templates should be applied, implicitly considering whether the resulting reaction would be both chemically feasible and strategic. We also propose an efficient hierarchical sampling to alleviate the computation cost. While achieving a significant improvement of $8.1\%$ over current state-of-the-art methods on the benchmark dataset, our model also offers interpretations for the prediction.

## 1 Introduction

Retrosynthesis planning is the procedure of identifying a series of reactions that lead to the synthesis of target product. It is first formalized by E. J. Corey [1] and now becomes one of the fundamental problems in organic chemistry. Such problem of "working backwards from the target" is challenging, due to the size of the search space–the vast numbers of theoretically-possible transformations–and thus requires the skill and creativity from experienced chemists. Recently, various computer algorithms [2] work in assistance to experienced chemists and save them tremendous time and effort.

The simplest formulation of retrosynthesis is to take the target product as input and predict possible reactants [1]. It is essentially the "reverse problem" of reaction prediction. In reaction prediction, the reactants (sometimes reagents as well) are given as the input and the desired outputs are possible products. In this case, atoms of desired products are the subset of reactants atoms, since the side products are often ignored (see Fig 1). Thus models are essentially designed to identify this subset in reactant atoms and reassemble them to be the product. This can be treated as a *deductive* reasoning process. In sharp contrast, retrosynthesis is to identify the superset of atoms in target products, and thus is an *abductive* reasoning process and requires "creativity" to be solved, making it a harder problem. Although recent advances in graph neural networks have led to superior performance in reaction prediction [3, 4, 5], such advances do not transfer to retrosynthesis.

Computer-aided retrosynthesis designs have been deployed over the past years since [6]. Some of them are completely rule-based systems [7] and do not scale well due to high computation cost and

---

[*]Work done while Hanjun was at Georgia Institute of Technology
[1]We will focus on this "single step" version of retrosynthesis in our paper.

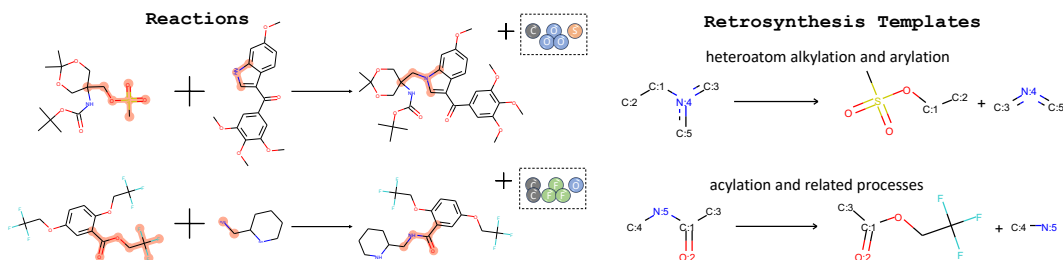

Figure 1: Chemical reactions and the retrosynthesis templates. The reaction centers are highlighted in each participant of the reaction. These centers are then extracted to form the corresponding template. Note that the atoms belong to the reaction side products (the dashed box in figure) are missing.

incomplete coverage of the rules, especially when rules are expert-defined and not algorithmically extracted [2]. Despite these limitations, they are very useful for encoding chemical transformations and easy to interpret. Based on this, the retrosim [8] uses molecule and reaction fingerprint similarities to select the rules to apply for retrosynthesis. Other approaches have used neural classification models for this selection task [9]. On the other hand, recently there have also been attempts to use the sequence-to-sequence model to directly predict SMILES [2] representation of reactants [10, 11] (and for the forward prediction problem, products [12, 13]). Albeit simple and expressive, these approaches ignore the rich chemistry knowledge and thus require huge amount of training. Also such models lack interpretable reasoning behind their predictions.

The current landscape of computer-aided synthesis planning motivated us to pursue an algorithm that shares the interpretability of template-based methods while taking advantage of the scalability and expressiveness of neural networks to learn when such rules apply. In this paper, we propose *Conditional Graph Logic Network* towards this direction, where chemistry knowledge about reaction templates are treated as logic rules and a conditional graphical model is introduced to tolerate the noise in these rules. In this model, the variables are molecules while the synthetic relationships to be inferred are defined among groups of molecules. Furthermore, to handle the potentially infinite number of possible molecule entities, we exploit the neural graph embedding in this model.

Our contribution can be summarized as follows:

1) We propose a new graphical model for the challenging retrosynthesis task. Our model brings both the benefit of the capacity from neural embeddings, and the interpretability from tight integration of probabilistic models and chemical rules.
2) We propose an efficient hierarchical sampling method for approximate learning by exploiting the structure of rules. Such algorithm not only makes the training feasible, but also provides interpretations for predictions.
3) Experiments on the benchmark datasets show a significant $8.1\%$ improvement over existing state-of-the-art methods in top-one accuracy.

**Other related work:** Recently there have been works using machine learning to enhance the rule systems. Most of them treat the rule selection as multi-class classification [9] or hierarchical classification [14] where similar rules are grouped into subcategories. One potential issue is that the model size grows with the number of rules. Our work directly models the conditional joint probability of both rules and the reactants using embeddings, where the model size is invariant to the rules.

On the other hand, researchers have tried to tackle the even harder problem of multi-step retrosynthesis [15, 16] using single-step retrosynthesis as a subroutine. So our improvement in single-step retrosynthesis could directly transfer into improvement of multi-step retrosynthesis [8].

## 2   Background

A chemical reaction can be seen as a transformation from set of $N$ reactant molecules $\{R_i\}_{i=1}^{N}$ to an outcome molecule $O$. Without loss of generality, we work with single-outcome reactions in this paper, as this is a standard formulation of the retrosynthetic problem and multi-outcome reactions can be split into multiple single-outcome ones. We refer to the set of atoms changed (e.g., bond being added or deleted) during the reaction as reaction centers. Given a reaction, the corresponding

retrosynthesis template $T$ is represented by a subgraph pattern rewriting rule [3]

$$T := o^T \to r_1^T + r_2^T + \ldots + r_{N(T)}^T, \tag{1}$$

where $N(\cdot)$ represents the number of reactant subgraphs in the template, as illustrated in Figure. 1. Generally we can treat the subgraph pattern $o^T$ as the extracted reaction center from $O$, and $r_i^T, i \in 1, 2, \ldots, N(T)$ as the corresponding pattern inside $i$-th reactant, though practically this will include neighboring structures of reaction centers as well.

We first introduce the notations to represent these chemical entities:

- **Subgraph patterns**: we use lower case letters to represent the subgraph patterns.
- **Molecule**: we use capital letters to represent the molecule graphs. By default, we use $O$ for an outcome molecule, and $R$ for a reactant molecule, or $M$ for any molecule in general.
- **Set**: sets are represented by calligraphic letters. We use $\mathcal{M}$ to denote the full set of possible molecules, $\mathcal{T}$ to denote all extracted retrosynthetic templates, and $\mathcal{F}$ to denote all the subgraph patterns that are involved in the known templates. We further use $\mathcal{F}_o$ to denote the subgraphs appearing in reaction outcomes, and $\mathcal{F}_r$ to denote those appearing in reactants, with $\mathcal{F} = \mathcal{F}_o \bigcup \mathcal{F}_r$.

**Task:** Given a production or target molecule $O$, the goal of a one-step retrosynthetic analysis is to identify a set of reactant molecules $\mathcal{R} \in \mathscr{P}(\mathcal{M})$ that can be used to synthesize the target $O$. Here $\mathscr{P}(\mathcal{M})$ is the power set of all molecules $\mathcal{M}$.

## 3 Conditional Graph Logic Network

Let $\mathbb{I}[m \subseteq M] : \mathcal{F} \times \mathcal{M} \mapsto \{0, 1\}$ be the predicate that indicates whether subgraph pattern $m$ is a subgraph inside molecule $M$. This can be checked via subgraph matching. Then the use of a retrosynthetic template $T : o^T \to r_1^T + r_2^T + \ldots + r_{N(T)}^T$ for reasoning about a reaction can be decomposed into two-step logic. First,

$$\text{I. Match template:} \quad \phi_O(T) := \mathbb{I}[o^T \subseteq O] \cdot \mathbb{I}[T \in \mathcal{T}], \tag{2}$$

where the subgraph pattern $o^T$ from the reaction template $T$ is matched against the product $O$, *i.e.*, $o^T$ is a subgraph of the product $O$. Second,

$$\text{II. Match reactants:} \quad \phi_{O,T}(\mathcal{R}) := \phi_O(T) \cdot \mathbb{I}[|\mathcal{R}| = N(T)] \cdot \prod_{i=1}^{N(T)} \mathbb{I}[r_i^T \subseteq R_{\pi(i)}], \tag{3}$$

where the set of subgraph patterns $\{r_1, \ldots, r_{N(T)}\}$ from the reaction template are matched against the set of reactants $\mathcal{R}$. The logic is that the size of the set of reactant $\mathcal{R}$ has to match the number of patterns in the reaction template $T$, and there exists a permutation $\pi(\cdot)$ of the elements in the reactant set $\mathcal{R}$ such that each reactant matches a corresponding subgraph pattern in the template.

Since there will still be uncertainty in whether the reaction is possible from a chemical perspective even when the template matches, we want to capture such uncertainty by allowing each template/or logic reasoning rule to have a different confidence score. More specifically, we will use a template score function $w_1(T, O)$ given the product $O$, and the reactant score function $w_2(\mathcal{R}, T, O)$ given the template $T$ and the product $O$. Thus the overall probabilistic models for the reaction template $T$ and the set of molecules $\mathcal{R}$ are designed as

$$\text{I. Match template:} \quad p(T|O) \propto \exp\left(w_1(T, O)\right) \cdot \phi_O(T), \tag{4}$$
$$\text{II. Match reactants:} \quad p(\mathcal{R}|T, O) \propto \exp\left(w_2(\mathcal{R}, T, O)\right) \cdot \phi_{O,T}(\mathcal{R}). \tag{5}$$

Given the above two step probabilistic reasoning models, the joint probability of a single-step retrosythetic proposal using reaction template $T$ and reactant set $\mathcal{R}$ can be written as

$$p(\mathcal{R}, T|O) \propto \exp\left(w_1(T, O) + w_2(\mathcal{R}, T, O)\right) \cdot \phi_O(T) \phi_{O,T}(\mathcal{R}), \tag{6}$$

In this energy-based model, whether the graphical model (GM) is directed or undirected is a design choice. We will present our directed GM design and the corresponding partition function in Sec 4 shortly. We name our model as *Conditional Graph Logic Network (GLN)* (Fig. 2), as it is a conditional graphical model defined with logic rules, where the logic variables are graph structures (*i.e.*, molecules, subgraph patterns, *etc.*). In this model, we assume that satisfying the templates is a necessary condition for the retrosynthesis, *i.e.*, $p(\mathcal{R}, T|O) \neq 0$ only if $\phi_O(T)$ and $\phi_{O,T}(\mathcal{R})$ are nonzero. Such restriction provides sparse structures into the model, and makes this abductive type of reasoning feasible.

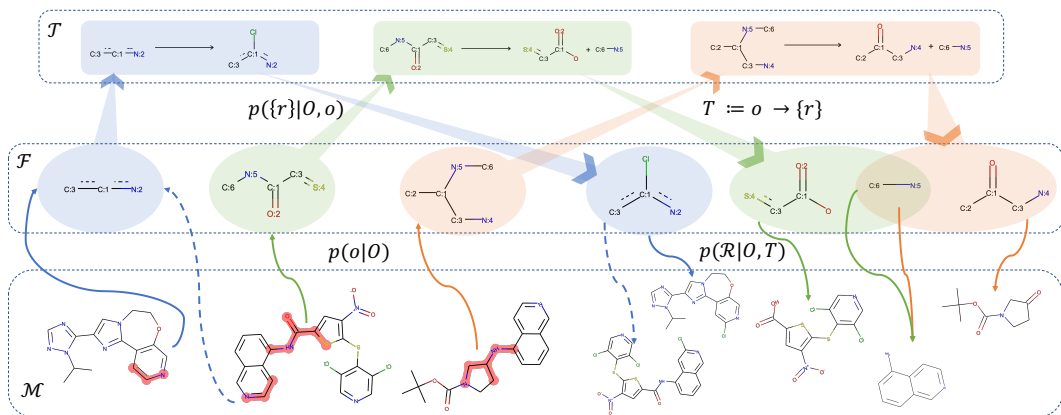

Figure 2: Retrosynthesis pipeline with GLN. The three dashed boxes from top to bottom represent set of templates $\mathcal{T}$, subgraphs $\mathcal{F}$ and molecules $\mathcal{M}$. Different colors represent retrosynthesis routes with different templates. The dashed lines represent potentially possible routes that are not observed. Reaction centers in products $O$ are highlighted.

**Reaction type conditional model:** In some situations when performing the retrosynthetic analysis, the human expert may already have a certain type $c$ of reaction in mind. In this case, our model can be easily adapted to incorporate this as well:

$$p(\mathcal{R}, T|O, c) \quad \propto \quad \exp\left(w_1\left(T, O\right) + w_2\left(\mathcal{R}, T, O\right)\right) \cdot \phi_O\left(T\right) \phi_{O,T}\left(\mathcal{R}\right) \mathbb{I}[T \in \mathcal{T}_c] \tag{7}$$

where $\mathcal{T}_c$ is the set of retrosynthesis templates that belong to reaction type $c$.

GLN is related but significantly different from Markov Logic Network (MLN, which also uses graphical model to model uncertainty in logic rules). MLN treats the predicates of logic rules as latent variables, and the inference task is to get the posterior for them. While in GLN, the task is the structured prediction, and the predicates are implemented with subgraph matching. We show more details on this connection in Appendix A.

## 4   Model Design

Although the model we defined so far has some nice properties, the design of the components plays a critical role in capturing the uncertainty in the retrosynthesis. We first describe a decomposable design of $p(T|O)$ in Sec. 4.1, for learning and sampling efficiency consideration; then in Sec. 4.2 we describe the parameterization of the scoring functions $w_1, w_2$ in detail.

### 4.1   Decomposable design of $p(T|O)$

Depending on how specific the reaction rules are, the template set $\mathcal{T}$ could be as large as the total number of reactions in extreme case. Thus directly model $p(T|O)$ can lead to difficulties in learning and inference. By revisiting the logic rule defined in Eq. (2), we can see the subgraph pattern $o^T$ plays a critical role in choosing the template. Since we represent the templates as $T = \left(o^T \rightarrow \left\{r_i^T\right\}_{i=1}^{N(T)}\right)$, it is natural to decompose the energy function $w_1(T, O)$ in Eq. (4) as $w_1(T, O) = v_1\left(o^T, O\right) + v_2\left(\left\{r_i^T\right\}_{i=1}^{N(T)}, O\right)$. Meanwhile, recall the template matching rule is also decomposable, so we obtain the resulting template probability model as:

$$p(T|O) = p(o^T, \left\{r_i^T\right\}_{i=1}^{N(T)} |O) \tag{8}$$
$$= \frac{1}{Z(O)}\left(\exp\left(v_1(o^T, O)\right) \cdot \mathbb{I}\left[o^T \in O\right]\right)\left(\exp\left(v_2\left(\left\{r_i^T\right\}_{i=1}^{N(T)}, O\right)\right) \cdot \mathbb{I}[(o^T \rightarrow \left\{r_i^T\right\}_{i=1}^{N(T)}) \in \mathcal{T}]\right),$$

where the partition function $Z(O)$ is defined as:

$$Z(O) = \sum_{o \in \mathcal{F}} \exp\left(v_1(o, O)\right) \cdot \mathbb{I}\left[o \in O\right] \cdot \left(\sum_{\{r\} \in \mathscr{P}(\mathcal{F})} \exp\left(v_2\left(\{r\}, O\right)\right) \cdot \mathbb{I}[(o \rightarrow \{r\}) \in \mathcal{T}]\right) \tag{9}$$

Here we abuse the notation a bit to denote the set of subgraph patterns as $\{r\}$.

With such decomposition, we can further speed up both the training and inference for $p(T|O)$, since the number of valid reaction centers per molecule and number of templates per reaction center are much smaller than total number of templates. Specifically, we can sample $T \sim p(T|O)$ by

first sampling reaction center $p(o|O) \propto \exp(v_1(o, O)) \cdot \mathbb{I}[o \in O]$ and then choosing the subgraph patterns for reactants $p(\{r\}|O, o) \propto \exp(v_2(\{r\}, O) \cdot \mathbb{I}[(o \to \{r\}) \in \mathcal{T}])$. In the end we obtain the templated represented as $(o \to \{r\})$.

In the literature there have been several attempts for modeling and learning $p(T|O)$, *e.g.*, multi-class classification [9] or multiscale model with human defined template hierarchy [14]. The proposed decomposable design follows the template specification naturally, and thus has nice graph structure parameterization and interpretation as will be covered in the next subsection.

Finally the directed graphical model design of Eq. (6) is written as

$$p(\mathcal{R}, T|O) = \frac{1}{Z(O)Z(T,O)} \exp\left(\left(v_1\left(o^T, O\right) + v_2\left(\left\{r_i^T\right\}_{i=1}^{N(T)}\right) + w_2(\mathcal{R}, T, O)\right)\right) \cdot \phi_O(T)\phi_{O,T}(\mathcal{R}) \quad (10)$$

where $Z(T, O) = \sum_{\mathcal{R} \in \mathscr{P}(\mathcal{M})} \exp(w_2(\mathcal{R}, T, O)) \cdot \phi_{O,T}(\mathcal{R})$ sums over all subsets of molecules.

## 4.2 Graph Neuralization for $v_1, v_2$ and $w_2$

Since the arguments of the energy functions $w_1, w_2$ are molecules, which can be represented by graphs, one natural choice is to design the parameterization based on the recent advances in graph neural networks (GNN) [17, 18, 19, 20, 21, 22]. Here we first present a brief review of the general form of GNNs, and then explain how we can utilize them to design the energy functions.

The graph embedding is a function $g : \mathcal{M} \bigcup \mathcal{F} \mapsto \mathbb{R}^d$ that maps a graph into $d$-dimensional vector. We denote $G = (\mathcal{V}^G, \mathcal{E}^G)$ as the graph representation of some molecule or subgraph pattern, where $\mathcal{V}^G = \{v_i\}_{i=1}^{|\mathcal{V}^G|}$ is the set of atoms (nodes) and $\mathcal{E}^G = \{e_i = (e_i^1, e_i^2)\}_{i=1}^{|\mathcal{E}^G|}$ is the set of bonds (edges). We represent each undirected bond as two directional edges. Generally, the embedding of the graph is computed through the node embeddings $h_{v_i}$ that are computed in an iterative fashion. Specifically, let $h_{v_i}^0 = x_{v_i}$ initially, where $x_{v_i}$ is a vector of node features, like the atomic number, aromaticity, *etc.* of the corresponding atom. Then the following update operator is applied recursively:

$$h_v^{l+1} = F(x_v, \{(h_u^l, x_{u \to v}\}_{u \in \mathcal{N}(v)}) \quad \text{where} \quad x_{u \to v} \text{ is the feature of edge } u \to v. \quad (11)$$

This procedure repeats for $L$ steps. While there are many design choices for the so-called message passing operator $F$, we use the structure2vec [21] due to its simplicity and efficient c++ binding with RDKit. Finally we have the parameterization

$$h_v^{l+1} = \sigma(\theta_1 x_v + \theta_2 \sum_{u \in \mathcal{N}(v)} h_u^l + \theta_3 \sum_{u \in \mathcal{N}(v)} \sigma(\theta_4 x_{u \to v})) \quad (12)$$

where $\sigma(\cdot)$ is some nonlinear activation function, *e.g.*, relu or tanh, and $\theta = \{\theta_1, \ldots, \theta_4\}$ are the learnable parameters. Let the node embedding $h_v = h_v^L$ be the last output of $F$, then the final graph embedding is obtained via averaging over node embeddings: $g(G) = \frac{1}{|\mathcal{V}^G|} \sum_{v \in \mathcal{V}^G} h_v$. Note that attention [23] or other order invariant aggregation can also be used for such aggregation.

With the knowledge of GNN, we introduce the concrete parametrization for each component:

• **Parameterizing $v_1$:** Given a molecule $O$, $v_1$ can be viewed as a scoring function of possible reaction centers inside $O$. Since the subgraph pattern $o$ is also a graph, we parameterize it with inner product, *i.e.*, $v_1(o, O) = g_1(o)^\top g_2(O)$. Such form can be treated as computing the compatibility between $o$ and $O$. Note that due to our design choice, $v_1(o, O)$ can be written as $v_1(o, O) = \sum_{v \in \mathcal{V}^O} h_v^\top g_1(o)$. Such form allows us to see the contribution of compatibility from each atom in $O$.

• **Parameterizing $v_2$:** The size of set of subgraph patterns $\{r_i^T\}_{i=1}^{N(T)}$ varies for different template $T$. Inspired by the DeepSet [24], we use average pooling over the embeddings of each subgraph pattern to represent this set. Specifically,

$$v_2(\{r_i^T\}_{i=1}^{N(T)}, O) = g_3(O)^\top \left(\frac{1}{N(T)} \sum_{i=1}^{N(T)} g_4(r_i^T)\right) \quad (13)$$

• **Parameterizing $w_2$:** This energy function also needs to take the set as input. Following the same design as $v_2$, we have

$$w_2(\mathcal{R}, T, O) = g_5(O)^\top \left(\frac{1}{|\mathcal{R}|} \sum_{R \in \mathcal{R}} g_6(R)\right). \quad (14)$$

Note that our GLN framework isn't limited to the specific parameterization above and is compatible with other parametrizations. For example, one can use condensed graph of reaction [25] to represent $\mathcal{R}$ as a single graph. Other chemistry specialized GNNs [3, 26] can also be easily applied here. For the ablation study on these design choices, please refer to Appendix C.1.

## 5  MLE with Efficient Inference

Given dataset $\mathcal{D} = \{(O_i, T_i, \mathcal{R}_i)\}_{i=1}^{|\mathcal{D}|}$ with $|\mathcal{D}|$ reactions, we denote the parameters in $w_1(T, O)$, $w_2(T, \mathcal{R}, O)$ as $\Theta = (\theta_1, \theta_2)$, respectively. The maximum log-likelihood estimation (MLE) is a natural choice for parameter estimation. Since $\forall (O, T, \mathcal{R}) \sim \mathcal{D}$, $\phi_O(T) = 1$ and $\phi_{O,T}(\mathcal{R}) = 1$, we have the MLE optimization as

$$\max_{\Theta} \ell(\Theta) \quad := \quad \widehat{\mathbb{E}}_{\mathcal{D}} [\log p(\mathcal{R}|T, O) p(T|O)] \tag{15}$$

$$= \quad \widehat{\mathbb{E}}_{\mathcal{D}} [w_1(T, O) + w_2(\mathcal{R}, T, O) - \log Z(O) - \log Z(O, T)],$$

The gradient of $\ell(\Theta)$ w.r.t. $\Theta$ can be derived[4] as

$$\nabla_{\Theta} \ell(\Theta) = \widehat{\mathbb{E}}_{\mathcal{D}} [\nabla_{\Theta} w_1(T, O)] - \widehat{\mathbb{E}}_O \mathbb{E}_{T|O} [\nabla_{\Theta} w_1(T, O)] \tag{16}$$

$$+ \widehat{\mathbb{E}}_{\mathcal{D}} [\nabla_{\Theta} w_2(\mathcal{R}, T, O)] - \widehat{\mathbb{E}}_{O,T} \mathbb{E}_{\mathcal{R}|T,O} [\nabla_{\Theta} w_2(\mathcal{R}, T, O)],$$

where $\mathbb{E}_{T|O}[\cdot]$ and $\mathbb{E}_{\mathcal{R}|O,T}[\cdot]$ stand for the expectation w.r.t. current model $p(T|O)$ and $p(\mathcal{R}, T|O)$, respectively. With the gradient estimator (16), we can apply the stochastic gradient descent (SGD) algorithm for optimizing (15).

**Efficient inference for gradient approximation:** Since $\mathcal{R} \in \mathscr{P}(\mathcal{M})$ is a combinatorial space, generally the expensive MCMC algorithm is required for sampling from $p(\mathcal{R}|T, O)$ to approximate (16). However, this can be largely accelerated by scrutinizing the logic property in the proposed model. Recall that the matching between template and reactants is the necessary condition for $p(\mathcal{R}, T|O) \geq 0$ by design. On the other hand, given $O$, only a few templates $T$ with reactants $\mathcal{R}$ have nonzero $\phi_O(T)$ and $\phi_{O,T}(\mathcal{R})$. Then, we can sample $T$ and $\mathcal{R}$ by importance sampling on *restricted supported templates* instead of MCMC over $\mathscr{P}(\mathcal{M})$. Rigorously, given $O$, we denote the matched templates as $\mathcal{T}_O$ and the matched reactants based on $T$ as $\mathcal{R}_{T,O}$, where

$$\mathcal{T}_O = \{T : \phi_O(T) \neq 0, \forall T \in \mathcal{T}\} \text{ and } \mathcal{R}_{T,O} = \{\mathcal{R} : \phi_{O,T}(\mathcal{R}) \neq 0, \forall \mathcal{R} \in \mathscr{P}(\mathcal{M})\} \tag{17}$$

Then, the importance sampling leads to an *unbiased* gradient approximation $\widehat{\nabla}_{\Theta} \ell(\Theta)$ as illustrated in Algorithm 1. To make the algorithm more efficient in practice, we have adopted the following accelerations:

- **1)** Decomposable modeling of $p(T|O)$ as described in Sec. 4.1;
- **2)** Cache the computed $\mathcal{T}_O$ and $\mathcal{R}(T, O)$ in advance.

In a dataset with $5 \times 10^4$ reactions, $|\mathcal{T}_O|$ is about 80 and $|\mathcal{R}_{T,O}|$ is roughly 10 on average. Therefore, we reduce the actual

---

**Algorithm 1** Importance Sampling for $\widehat{\nabla}_{\Theta} \ell(\Theta)$

1: Input $(\mathcal{R}, T, O) \sim \mathcal{D}$, $p(\mathcal{R}|T, O)$ and $p(T|O)$.
2: Construct $\mathcal{T}_O$ according to $\phi_O(T)$.
3: Sample $\tilde{T} \propto \exp(w_1(T, O))$, $\forall T \in \mathcal{T}_O$ in hierarchical way, as in Sec. 4.1.
4: Construct $\mathcal{R}_{T,O}$ according to $\phi_{O,T}(\mathcal{R})$.
5: Sample $\tilde{\mathcal{R}} \propto \exp(w_2(\mathcal{R}, T, O))$.
6: Compute stochastic approximation $\widehat{\nabla}_{\Theta} \ell(\Theta)$ with sample $\left(\mathcal{R}, T, \tilde{\mathcal{R}}, \tilde{T}, O\right)$ by (16).

---

computational cost to a manageable constant. We further reduce the computation cost of sampling by generating the $T$ and $\mathcal{R}$ uniformly from the support. Although these samples only cover the support of the model, we avoid the calculation of the forward pass of neural networks, achieving better computational complexity. In our experiment, such an approximation already achieves state-of-the-art results. We would expect recent advances in energy based models would further boost the performance, which we leave as future work to investigate.

**Remark on $\mathcal{R}_{T,O}$:** Note that to get all possible sets of reactants that match the reaction template $T$ and product $O$, we can efficiently use graph edit tools without limiting the reactants to be known in the dataset. This procedure works as follows: given a template $T = o^T \to r_1^T + \ldots + r_N^T$,

1) Enumerate all matches between subgraph pattern $o^T$ and target product $O$.
2) Instantiate a copy of the reactant atoms according to $r_1^T, \ldots, r_N^T$ for each match.
3) Copy over all of the connected atoms and atom properties from $O$.

This process is a routine in most Cheminformatics packages. In our paper we use `runReactants` from RDKit with the improvement of stereochemistry handling [5] to realize this.

**Further acceleration via beam search:** Given a product $O$, the prediction involves finding the pair $(\mathcal{R}, T)$ that maximizes $p(\mathcal{R}, T|O)$. One possibility is to first enumerate $T \in \mathcal{T}(O)$ and then $\mathcal{R} \in \mathcal{R}_{T,O}$. This is acceptable by exploiting the sparse support property induced by logic rules.

A more efficient way is to use beam search with size $k$. Firstly we find $k$ reaction centers $\{o_i\}_{i=1}^k$ with top $v_1(o, O)$. Next for each $o \in \{o_i\}_{i=1}^k$ we score the corresponding $v_2(\{r\}, O) \cdot \mathbb{I}\left[(o \rightarrow \{r\}) \in \mathcal{T}\right]$. In this stage the top $k$ pairs $\{(o_{T_j}, \{r_i^{T_j}\})\}_{j=1}^k$ (*i.e.*, the templates) that maximize $v_1(o|O) + v_2(\{r\}, O)$ are kept. Finally using these templates, we choose the best $\mathcal{R} \in \bigcup_{j=1}^k \mathcal{R}_{T_j, O}$ that maximizes total score $w_1(T, O) + w_2(\mathcal{R}, T, O)$. Fig. 2 provides a visual explanation.

# 6 Experiment

**Dataset:** We mainly evaluate our method on a benchmark dataset named USPTO-50k, which contains 50k reactions of 10 different types in the US patent literature. We use exactly the same training/validation/test splits as Coley et al. [8], which contain 80%/10%/10% of the total 50k reactions. Table 1 contains the detailed information about the benchmark. Additionally, we also build a dataset from the entire USPTO 1976-2016 to verify the scalability of our method.

**Baselines:** Baseline algorithms consist of rule-based ones and neural network-based ones, or both. The expertSys is an expert system based on retrosynthetic reaction rules, where the rule is selected according to the popularity of the corresponding reaction type. The seq2seq [10] and transformer [11] are neural sequence-to-sequence-based learning model [28] implemented with LSTM [29] or Transformer [30]. These models encode the canonicalized SMILES representation of the target compound as input, and directly output canonical SMILES of reactants. We also include some data-driven template-based models. The retrosim [8] uses direct calculation of molecular similarities to rank the rules and resulting reactants. The neuralsym [9] models $p(T|O)$ as multi-class classification using MLP. All the results except neuralsym are obtained from their original reports, since we have the same experiment setting. Since neuralsym is not open-source, we reimplemented it using their best reported ELU512 model with the same method for parameter tuning.

**Evaluation metric:** The evaluation metric we used is the top-$k$ exact match accuracy, which is commonly used in the literature. This metric compares whether the predicted set of reactants are *exactly* the same as ground truth reactants. The comparison is performed between canonical SMILES strings generated by RDKit.

**Setup of GLN:** We use rdchiral [31] to extract the retrosynthesis templates from the training set. After removing duplicates, we obtained 11,647 unique template rules in total for USPTO-50k. These rules represent 93.3% coverage of the test set. That is to say, for each test instance we try to apply these rules and see if any of the rules gives exact match. Thus this is the theoretical upper bound of the rule-based approach using this particular degree of specificity, which is high enough for now. For more information about the statistics of these rules, please refer to Table 2.

We train our model for up to 150k updates with batch size of 64. It takes about 12 hours to train with a single GTX 1080Ti GPU. We tune embedding sizes in $\{128, 256\}$, GNN layers $\{3, 4, 5\}$ and GNN aggregation in $\{\texttt{max}, \texttt{mean}, \texttt{sum}\}$ using validation set. Our code is released at https://github.com/Hanjun-Dai/GLN. More details are included in Appendix B.

## 6.1 Main results

We present the top-$k$ exact match accuracy in Table 3, where $k$ ranges from $\{1, 3, 5, 10, 20, 50\}$. We evaluate both the reaction class unknown and class conditional settings. Using the reaction class as prior knowledge represents some situations where the chemists already have an idea of how they would like to synthesize the product.

In all settings, our proposed GLN outperforms the baseline algorithms. And particularly for top-1 accuracy, our model performs significantly better than the second best method, with 8.1% higher accuracy with unknown reaction class, and 8.9% higher with reaction class given. This demonstrates the advantage of our method in this difficult setting and potential applicability in reality.

| USPTO 50k | |
| --- | --- |
| # train | 40,008 |
| # val | 5,001 |
| # test | 5,007 |
| # rules | 11,647 |
| # reaction types | 10 |

Table 1: Dataset information.

| | |
| --- | --- |
| Rule coverage | 93.3% |
| # unique centers | 9,078 |
| Avg. # centers per mol | 29.31 |
| Avg. # rules per mol | 83.85 |
| Avg. # reactants | 1.71 |

Table 2: Reaction and template set information.

| | Top-$k$ accuracy % | | | | | |
| --- | --- | --- | --- | --- | --- | --- |
| methods | 1 | 3 | 5 | 10 | 20 | 50 |
| Reaction class unknown | | | | | | |
| transformer[11] | 37.9 | 57.3 | 62.7 | / | / | / |
| retrosim[8] | 37.3 | 54.7 | 63.3 | 74.1 | 82.0 | 85.3 |
| neuralsym[9] | 44.4 | 65.3 | 72.4 | 78.9 | 82.2 | 83.1 |
| GLN | **52.5** | **69.0** | **75.6** | **83.7** | **89.0** | **92.4** |
| Reaction class given as prior | | | | | | |
| expertSys[10] | 35.4 | 52.3 | 59.1 | 65.1 | 68.6 | 69.5 |
| seq2seq[10] | 37.4 | 52.4 | 57.0 | 61.7 | 65.9 | 70.7 |
| retrosim[8] | 52.9 | 73.8 | 81.2 | 88.1 | 91.8 | 92.9 |
| neuralsym[9] | 55.3 | 76.0 | 81.4 | 85.1 | 86.5 | 86.9 |
| GLN | **64.2** | **79.1** | **85.2** | **90.0** | **92.3** | **93.2** |

Table 3: Top-$k$ exact match accuracy.

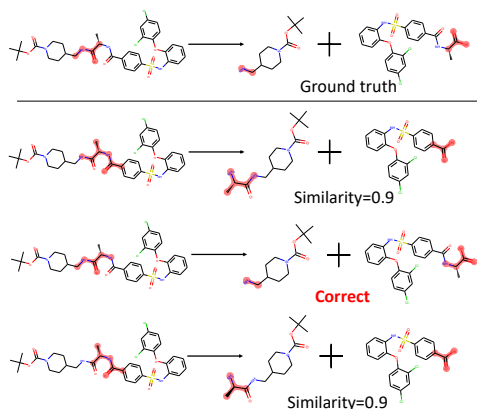

Figure 3: Example successful predictions.

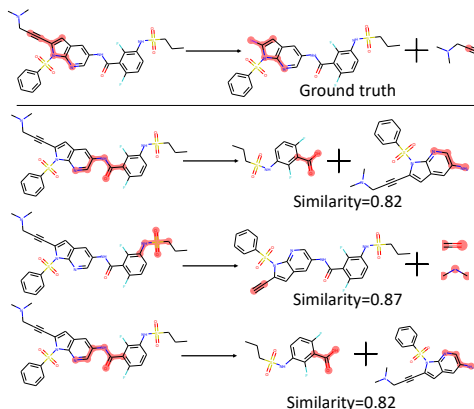

Figure 4: Example failed predictions.

Moreover, our performance in the reaction class unknown setting even outperforms expertSys and seq2seq in the reaction conditional setting. Since the transformer paper didn't report top-$k$ performance for $k > 10$, we leave it as blank. Meanwhile, Karpov et al. [11] also reports the result when training using training+validation set and tuning on the test set. With this extra priviledge, the top-1 accuracy of transformer is 42.7% which is still worse than our performance. This shows the benefit of our logic powered deep neural network model comparing to purely neural models, especially when the amount of data is limited.

Since the theoretical upper bound of this rule-based implementation is 93.3%, the top-50 accuracy for our method in each setting is quite close to this limit. This shows the probabilistic model we built matches the actual retrosynthesis target well.

## 6.2 Interpret the predictions

**Visualizing the predicted synthesis:** In Fig 3 and 4, we visualize the ground truth reaction and the top 3 predicted reactions (see Appendix C.6 for high resolution figures). For each reaction, we also highlight the corresponding reaction cores (*i.e.*, the set of atoms get changed). This is done by matching the subgraphs from predicted retrosynthesis template with the target compound and generated reactants, respectively. Fig 3 shows that our correct prediction also gets almost the same reaction cores predicted as the ground truth. In this particular case, the explanation of our prediction aligns with the existing reaction knowledge.

Fig 4 shows a failure mode where none of the top-3 prediction matches. In this case we calculated the similarity between predicted reactants and ground truth ones using Dice similarity from RDKit. We find these are still similar in the molecule fingerprint level, which suggests that these predictions could be the potentially valid but unknown ones in the literature.

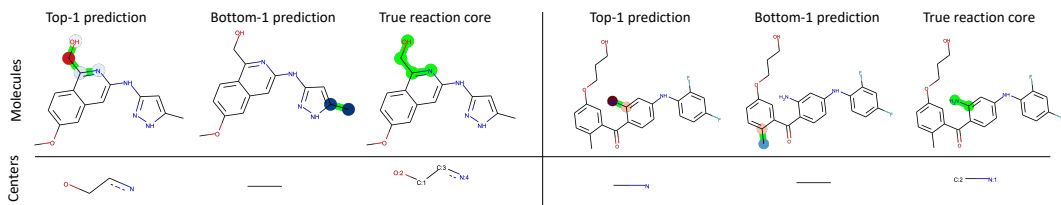

Figure 5: Reaction center prediction visualization. Red atoms indicate positive match scores, while blue ones having negative scores. The darkness of the color shows the magnitude of the score. Green parts highlight the substructure match between molecules and center structures.

**Visualizing the reaction center prediction:** Here we visualize the prediction of probabilistic modeling of reaction center. This is done by calculating the inner product of each atom embedding in target molecule with the subgraph pattern embedding. Fig 5 shows the visualization of scores on the atoms that are part of the reaction center. The top-1 prediction assigns positive scores to these atoms (red ones), while the bottom-1 prediction (*i.e.*, prediction with least probability) assigns large negative scores (blue ones). Note that although the reaction center in molecule and the corresponding subgraph pattern have the same structure, the matching scores differ a lot. This suggests that the model has learned to predict the activity of substructures inside molecule graphs.

## 6.3 Study of the performance

In addition to the overall numbers in Table 3, we provide detailed study of the performances. This includes per-category performance, the accuracy of each module in hierarchical sampling and also the effect of the beam size. Due to the space limit, please refer to Appendix C.

## 6.4 Large scale experiments on USPTO-full

To see how this method scales up with the dataset size, we create a large dataset from the entire set of reactions from USPTO 1976-2016. There are 1,808,937 raw reactions in total. For the reactions

|        | retrosim | neuralsym | GLN  |
|--------|----------|-----------|------|
| top-1  | 32.8     | 35.8      | **39.3** |
| top-10 | 56.1     | 60.8      | **63.7** |

Table 4: Top-k accuracy on USPTO-full.

with multiple products, we duplicate them into multiple ones with one product each. After removing the duplications and reactions with wrong atom mappings, we obtain roughly 1M unique reactions, which are further divided into train/valid/test sets with size 800k/100k/100k.

We train on single GPU for 3 days and report with the model having best validation accuracy. The results are presented in Table 4. We compare with the best two baselines from previous sections. Despite the noisiness of the full USPTO set relative to the clean USPTO-50k, our method still outperforms the two best baselines in top-$k$ accuracies.

# 7 Discussion

**Evaluation:** Retrosynthesis usually does not have a single right answer. Evaluation in this work is to reproduce what is reported for single-step retrosynthesis. This is a good, but imperfect benchmark, since there are potentially many reasonable ways to synthesize a single product.

**Limitations:** We share the limitations of all template-based methods. In our method, the template designs, more specifically, their specificities, remain as a design art and are hard to decide beforehand. Also, the scalability is still an issue since we rely on subgraph isomorphism during preprocessing.

**Future work:** The subgraph isomorphism part can potentially be replaced with predictive model, while during inference the fast inner product search [32] can be used to reduce computation cost. Also actively building templates or even inducing new ones could enhance the capacity and robustness.

## Acknowledgments

We would like to thank anonymous reviewers for providing constructive feedbacks. This project was supported in part by NSF grants CDS&E-1900017 D3SC, CCF-1836936 FMitF, IIS-1841351, CAREER IIS-1350983 to L.S.

## Footnotes

[2]https://www.daylight.com/dayhtml/doc/theory/theory.smiles.html.

[3] Commonly encoded using SMARTS/SMIRKS patterns

[4]We adopt the conventions $0 \log 0 = 0$ [27], which is justified by continuity since $x \log x \to 0$ as $x \to 0$.

[5]https://github.com/connorcoley/rdchiral.

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
