[Supplementary Material · app.pdf]

# Appendix

## A    Connection to Markov Logic Network

Similar to the proposed model, the Markov logic network (MLN) [33] is an alternative to introduce uncertainty into logic rules. However, there is significant difference in the way the retrosynthetic templates are treated. The proposed model considers the templates as separate variables that will be inferred for the target molecules together with the reactions. The explicit probabilistic modeling of templates makes it more straightforward to interpret the prediction. The MLN instead sets the logic rules (the templates) as features in the energy-based model, *i.e.*, $p(\mathcal{R}|O) \propto \exp\left(\sum_{T \in \mathcal{T}} w_{T,O}\phi_O(T) + w_{T,\mathcal{R},O}\phi_{O,T}(\mathcal{R})\right)$, upon which the template inference is not well-defined. Moreover, our model will also lead to efficient sampling and inference, avoiding the MCMC on combinatorial space $\mathscr{P}(\mathcal{M})$ in the MLN, which accelerates the model learning.

So in summary:

- GLN is a directed graphical model while MLN is undirected.
- MLN treats the predicates of logic rules as latent variables, and the inference task is to get the posterior of them. While in GLN, the task is the structured prediction, and the predicates are implemented with subgraph matching.
- Due to the above two, GLN can be implemented with efficient hierarchical sampling. However for MLN, generally the expensive MCMC in combinatorial space is needed for both training and inference.

## B    Details of setup

Figure 6: Distribution of reaction types.

**Dataset information**    Figure 6 shows the distribution of reactions over 10 types. We can see this dataset is highly unbalanced.

**Implementation details**    The preprocessing of $\mathcal{T}_O$ and $\mathcal{R}_{T,O}$ is relatively expensive, since theoretically the subgraph isomorphism check is NP-hard. However, since the processing is embarrassingly parallelizable, it took about 1 hour on a cluster with 48 CPU cores for 50k reactions.

We implement the entire model using pytorch. The optimizer we used is Adam [34] with a fixed learning rate of $1e-3$ and a gradient clip of $5.0$.

In all the experiments, the graph embedding module is implemented using s2v [21]. The best embedding size we used has size of 256 for representing each molecule or subgraph structure, and `relu` is used as nonlinear activation function.

For the aggregation used in $g(\cdot)$, in DeepSet module used for representation of $r_i^{T} \, {}_{i=1}^{N(T)}$ for a specific $T$, or in DeepSet module for molecule set $\mathcal{R}$, we tried {max, sum, average}-pooling, and found the performance is about the same. We use $average$-pooling since it offers the scoring of each node embedding within the graphs. The visualization in Fig 5 relies on this trick.

|         | s2v-3 | GGNN | MPNN | GIN  | ECFP | s2v-0 | s2v-1 | s2v-2 |
|---------|-------|------|------|------|------|-------|-------|-------|
| top-1   | 52.6  | 51.6 | 50.4 | 51.8 | 51.9 | 40.7  | 47.0  | 51.3  |
| top-10  | 83.1  | 81.8 | 83.2 | 83.3 | 81.5 | 78.1  | 80.4  | 82.2  |

Table 5: Ablation study on USPTO-50k with different representations.

| Class | Fraction % |
|-------|------------|
| 1     | 30.3       |
| 2     | 23.8       |
| 3     | 11.3       |
| 4     | 1.8        |
| 5     | 1.3        |
| 6     | 16.5       |
| 7     | 9.2        |
| 8     | 1.6        |
| 9     | 3.7        |
| 10    | 0.5        |

Figure 7: Reaction distribution over 10 types.

Figure 8: Top-10 accuracy per each reaction type.

## C  More experiment results

### C.1  Ablation study of design choices

Our GLN provides a general graphical model to retrosynthesis problem, which is compatible with many reasonable choices of the representation of graphs. In addition to `structure2vec` with 3 layers (s2v-3) we used in the paper, we provide more ablation studies using different widely used GNNs and different number of "message-passing" layers.

The rationale behind the choices are: 1) the GNNs should be able to take both atom and bond features into consideration; 2) according to Xu et al. [35], the family of message-passing GNNs should have similar representation power as WL graph isomorphism check at best. We adopt the s2v in our paper since it satisfies these requirements. Meanwhile, it comes with efficient `c++` binding of `RDKit`.

We use 2 layers of GNN by default, or use -$k$ after the name in Table 5 to denote $k$-layer design. We can see that most variations of GNNs can achieve similar performances with enough number of message-passing like propagations. Based on this, for the experiment on the full USPTO dataset we simply use ECFP-2 provided by RDKit, as it is WL-isomorphism check based method with enough expressiveness [35] but faster to run.

Besides the choice of GNN, we also compare the choices of $v_1$, $v_2$ and $w_2$ mentioned in Section 4.2. Basically all these functions are comparing the compatibility of two vectors $\vec{x}, \vec{y}$. In the paper, we simply used inner-product $\vec{x}^{\top}\vec{y}$. Here we also studied $MLP([\vec{x}, \vec{y}])$ and bilinear $\vec{x}^{\top}A\vec{y}$. For top-1, the inner-prod, MLP and bilinear gets 52.6, 52.7 and 53.5, respectively. So our GLN could be further improved with better design choices.

### C.2  Per-category performance

We study the performance per each reaction category. Following the setting of baseline methods, we report the top-10 accuracy. As is shown in Table 7, the distribution of reaction types is highly unbalanced. From Fig 8 we can see our performances are better than retrosim in most classes, including the most common cases like class 1 and 2, or rare cases like class 4 or 8. This shows that our performance is not obtained by overfitting to one particular category of reactions. Such property is also important, as the retrosynthesis could involve rare reactions that haven't been well studied in the literature.

For per category performance for reaction type conditional tasks, as well the effect of beam-size, please refer to Appendix C.

### C.3  Reaction conditional performance

In Figure 9 we show the per-class performance when the reaction type is given as prior. As is shown Figure 6, the distribution of reaction types is not uniform, where some reactions only get less than 5%

Figure 9: Top-10 accuracy per reaction class, when the reaction class is given during training.

Figure 10: Top-$k$ accuracy with different beam sizes.

Figure 11: Inference speed with different beam sizes.

Figure 12: Top-$k$ accuracy of reaction center and template.

of the total data. In this case, it is important to have a flexible model that can take the reaction type into account. Training one model per each reaction class is not a good idea in this case due to the imbalance of distribution.

From Figure 9 we can see our performances are comparable to retrosim in all classes, while being much better than expertSys and seq2seq. Even in rare classes like class 9 or 10, we can still get best or second best performance. This shows the effectiveness and the flexibility of our GLN.

### C.4  Effect of beam size

**Beam size** In Section. 6.1 we reported the top-$k$ accuracy with beam size of 50, since $k$ is at most 50. Here we study the performance of GLN using different beam sizes. Figure 10 shows the top-$k$ accuracy for different $k$ and different beam sizes. Overall the performance gets consistently better with larger beam sizes, for all top-$k$ predictions. We can also see that the top-1 accuracy improved about 10% from beam size 1 (*i.e.*, greedy inference) to beam size 3. Note that the curve of beam size $s$ flattened after top-$s$ predictions, since generally it didn't produce more predictions than $s$.

We also report the speed for inference in Figure 11. Such information during inference is averaged over 5,007 test predictions. The majority of the time is spent during applying the template via the call to RDKit, thus the time required grows up linearly with the beam size, as the number of RDKit calls grows linearly with the beam size.

**Accuracy of** $p(T|O)$ In Figure 12 we show the accuracy of $p(T|O)$, which decomposes into the reaction center identification accuracy and the template selection accuracy related to that reaction center. Here the beam size is fixed to 50. Predicting the reaction center is relatively easy and GLN achieves 99% top-20 accuracy. These results indicate that the current bottleneck in performance is in the template selection part, which is reasonably good now but can definitely be further improved by capturing more reaction features.

### C.5  Generalize logic check $\phi_O(T)$

The logic function $\phi_O(T)$ comes with our GLN can be potentially applied to any rule based systems. For example, when combined with neuralsym [9] (we denote the modified one as $\phi$-neuralsym), it

Figure 13: Example successful predictions.

Figure 14: Example failed predictions.

further reduces the space of template selection. $\phi$-neuralsym gets top-1 accuracy of 46.9% and 57.7% in reaction type unconditional and conditional cases, respectively. This is about 2% improvement over its vanilla performance.

## C.6 Visualization results

In Figure 13 and 13 we put examples of successful and failed predictions with better resolution.