[Reviews · NeurIPS 2019]

Reviewer 1



The paper proposes a graph neural network based method to solve the retrosynthesis prediction problem, that is the identification of the reactions which lead to a particular target. In particular, the proposed graphical model exploit ideas from graph neural networks, in order to learn informative representation. Furthermore, expert knowledge from chemistry rules can be integrated in order to consider known restrictions and to provide interpretable solutions. The paper well describes the background and task of interest. However, I find the notation and model derivations to be hard to follow. The results show very promising performances, but could be more comprehensive and I find the methodological contribution to be limited. Detailed comments are provided below. 1) Overall, I found the model design choices to be not properly justified. I would suggest to clarify the arguments and formula derivations in section 4, in order to make the text smoother and easier to follow. 2) I would also recommend to properly introduce the notation and definitions before presenting the model. 3) Graph Neuralization. The design and model choices seem to be arbitrary. Could the author(s) discuss further on it? 4) Since the paper seems to be heavily focused on the application and the strong experimental results, I would suggest to explore additional options for the update operator of the graph neural network (additionally to structure2vec). This would validate the actual benefit of the learning component. 5) Using only one dataset is restrictive in order to assess the generalisability of the method. I would consider to validate the model on additional real and/or simulated datasets %%%%%%%%%%%%%%% Thank you for your rebuttal and for answering my concerns. I found the new experiments and arguments provided to be convincing.

Reviewer 2



I really enjoyed reading this paper - it is well motivated and the theory is also well described in my opinion. Originality: The combination of merging a rule-based approach with a neural approach appears to be very well suited for the problem at hand. It has to the best of my knowledge not been studied using graphical models, exploiting the compositional structure of these rules, which is a significant step forward. The compositionality of the rules has been studied in a purely symbolic way by Segler et al in https://onlinelibrary.wiley.com/doi/abs/10.1002/chem.201604556 which I would suggest to additionally cite. Approaching the problem by decomposing the the rule application into several matching and reaction scoring steps is very elegant. Do the authors think the choice of the graph neural network for the encoding plays a large role? ___________ EDIT after authors response: I want to thank the authors for addressing the questions. With the additional data provided during the rebuttal (which should be added to the final version), I raise my score accordingly.

Reviewer 3



Positives: The paper is well organized, with each section clearly defined and good use of notation to clearly mark research objectives and contributions made by the authors. The introduction sets up the contributions clearly, and the background/method sections manage to cover a lot of material with varying degrees of success. The figures/graphics provided by the paper also do a good job of expressing what the machine learning task that is being solved is and the proposed solution as it relates to retrosynthesis. The authors focus on a specific ML task, retrosynthesis, is also refreshing as it’s applications in the industry are clear. The mathematical equations provide a means to implement the model as well, this also extends to descriptions for the model including layers and optimization functions. In the experiment section, the authors model also appears to beat out the baselines. Negatives: The paper would have benefited with a pseudo-code algorithm describing the graph neutralization for v1, v2 and w2. The equations, while well written, are frequently written inline and the clarity of the overall algorithm is a little unclear. Overall the focus on retrosynthesis as the main ML task leads to equations being described from that perspective solely. It would have helped if the authors described the methods agnostic of the data or ML task. The related work section is also fairly short with citations to other works infrequently mentioned in the paper. In particular, the paper mentions markov logic networks, which seem structurally similar to the proposed GLNs. The authors mention an appendix A which explains the differences, but this was not available for review. Given the strong similarities between the two frameworks, GLNs novelty is called into question. It would seem this is just an application of MLNs in the context of retrosynthesis. Related work: The paper provides a related work section, which very quickly mentions approaches dealing with the second goal of the paper, hierarchical classification. The authors do a well enough job citing previous methods which they base their model off of. The baseline methods are also cited and explained. Conclusion: The following paper does a good job describing the model proposed, and the solution does appear to improve upon previous approaches dealing with retrosynthesis. The authors focus on a specific ML task is also a step in the right direction, however their focus in describing the model in regards to the task makes applying the model to other ML tasks difficult. The use of inline equations also removes some clarity in their approach. The novelty of the approach is also called into question since there are clear similarities with MLNs, and the appendix was not available. Given that, I would give this paper a weak reject as it is unclear if this is simply an application paper or a truly new method.

[Author Response · NeurIPS 2019]

We thank the reviewers for their insightful comments, which we will incorporate into the revised version. We first
address some common concerns raised by the reviewers by providing additional experiment results.

**Design choices of graph neural networks (GNN)**

Our GLN provides a general graphical model to retrosynthesis problem, which is compatible with many reasonable
choices of the representation of graphs. In addition to `structure2vec` with 3 layers (s2v-3) we used in the paper, we
provide more ablation studies using different widely used GNNs and different number of "message-passing" layers.

| | s2v-3 | GGNN | MPNN | GIN | ECFP | s2v-0 | s2v-1 | s2v-2 | | retrosim | neuralsym | GLN |
|---|---|---|---|---|---|---|---|---|---|---|---|---|
| top-1 | 52.6 | 51.6 | 50.4 | 51.8 | 51.9 | 40.7 | 47.0 | 51.3 | top-1 | 32.8 | 35.8 | **39.3** |
| top-10 | 83.1 | 81.8 | 83.2 | 83.3 | 81.5 | 78.1 | 80.4 | 82.2 | top-10 | 56.1 | 60.8 | **63.7** |

Table 1: Ablation study on schneider-50k with different representations.    Table 2: Top-k accuracy on USPTO-all.

The rationale behind the choices are: 1) the GNNs should be able to take both atom and bond features into consideration;
2) according to [1], the family of message-passing GNNs should have similar representation power as WL graph
isomorphism check at best. We adopt the s2v in our paper since it satisfies these requirements. Meanwhile, it comes
with efficient `c++` binding of `RDKit`. We will elaborate on the details in our revision.

We use 2 layers of GNN by default, or use -$k$ after the name in Table 1 to denote $k$-layer design. We can see that most
variations of GNNs can achieve similar performances with enough number of message-passing like propagations.

**Reviewer 1** **Q1: validate the model on additional real and/or simulated datasets:** We added experiments on a
larger USPTO dataset of $\sim$1M reactions (after expanding multi-product reactions and removing duplicates), divided
into train/valid/test sets of 800k/100k/100k. The results are presented in Table 2. Despite the noisiness of the full
USPTO set relative to the schneider-50k subset, our method still outperforms the two best baselines in top-$k$ accuracies.

**Q2: design choices in section 4:** Besides the choice of GNN in Table 1, we also compare the choices of $v_1$, $v_2$ and
$w_2$. Basically all these functions are comparing the compatibility of two vectors $\vec{x}, \vec{y}$. In the paper, we simply used
inner-product $\vec{x}^\top \vec{y}$. Here we also studied $MLP([\vec{x}, \vec{y}])$ and bilinear $\vec{x}^\top A \vec{y}$. For top-1, the inner-prod, MLP and bilinear
gets 52.6, 52.7 and 53.5, respectively. So our GLN could be further improved with better design choices.

We emphasize that the proposed GLN is general enough which is compatible with other parametrizations. We will
make notations and definitions clearer in the paper, and also refine the text for readability.

**Reviewer 2** **Q1: where is the origin of the reactant set $\mathcal{R}$:**

We first clarify that we don't limit the reactants to be known in training/test set; this method is able to plan multi-step
synthetic routes by recursively making single-step predictions. As in Eq (5), we are modeling the distribution of $\mathcal{R}$ given
template $T$ and target product $O$. With the help of $T = o^T \to r_1^T + \ldots + r_N^T$, we can generate all possible reactants in
entire molecule space that satisfy the logic rules. This procedure works as follows: **1)** Enumerate all matches between
subgraph pattern $o^T$ and target product $O$. **2)** Instantiate a copy of the reactant atoms according to $r_1^T, \ldots, r_N^T$ for each
match. **3)** Copy over all of the connected atoms and atom properties from $O$.

This process is a routine in most cheminformatics packages (we use RDKit's `runReactants`). As mentioned in Line
209 in main paper, empirically the # candidate sets is 10 on average, which is not so expensive.

**Reviewer 3**: We would first clarify that the appendix file was submitted as part of the supplementary.

**Q1: comparison with MLN:** We briefly summarize here, further details can be found in Appendix A. While our GLN
and MLN are related–both combine logic rules with graphical models–there are some significant differences:

• GLN is a directed graphical model while MLN is undirected.
• MLN treats the predicates of logic rules as latent variables, and the inference task is to get the posterior of them.
While in GLN, the task is the structured prediction, and the predicates are implemented with subgraph matching.
• Due to the above two, GLN can be implemented with efficient hierarchical sampling. However for MLN, generally
the expensive MCMC in combinatorial space is needed for both training and inference.

**Q2: "better to have task agnostic desc...unclear if this is simply an application paper or a truly new method."**

We are targeting a fundamental problem in ML community, *i.e.*, how to learn distributions on combinatorial discrete
variables. The proposed GLN exploits the logic rules to avoid the expensive MCMC sampling over the combinatorial
space whose mixing time can be extremely slow. We used the chemical retrosynthesis as our task of focus, which is
challenging due to the exact the reason that the modeling objects are discrete and combinatorial. Meanwhile, this task is
urgent in need and has been attracting more attentions recently both in ML community and chemistry.

We will fix the inline equations and refine the manuscript based on your suggestions.

[1] *Xu et.al*, How Powerful are Graph Neural Networks? *ICLR 2019*

[Meta-Review · NeurIPS 2019]

As the reviewers' concerns were clarified in the author feedback, they unanimously recommend to accept the paper.